# Differential Responses of Retinal Neurons and Glia Revealed via Proteomic Analysis on Primary and Secondary Retinal Ganglion Cell Degeneration

**DOI:** 10.3390/ijms241512109

**Published:** 2023-07-28

**Authors:** Jacky M. K. Kwong, Joseph Caprioli, Joanne C. Y. Lee, Yifan Song, Feng-Juan Yu, Jingfang Bian, Ying-Hon Sze, King-Kit Li, Chi-Wai Do, Chi-Ho To, Thomas Chuen Lam

**Affiliations:** 1Ophthalmology, Stein Eye Institute, University of California Los Angeles, Los Angeles, CA 90095, USA; caprioli@jsei.ucla.edu (J.C.); joannelee426@g.ucla.edu (J.C.Y.L.); dr.songyifan@foxmail.com (Y.S.); 2Centre for Myopia Research, School of Optometry, The Hong Kong Polytechnic University, Hong Kong, China; jessicayfj.yu@polyu.edu.hk (F.-J.Y.); jingfang.jf.bian@polyu.edu.hk (J.B.); 18074319r@connect.polyu.hk (Y.-H.S.); kk.li@polyu.edu.hk (K.-K.L.); chi-wai.do@polyu.edu.hk (C.-W.D.); chi-ho.to@polyu.edu.hk (C.-H.T.); 3Research Centre for SHARP Vision (RCSV), The Hong Kong Polytechnic University, Hong Kong, China; 4Centre for Eye and Vision Research (CEVR), The Hong Kong Polytechnic University, 17W, Hong Kong Science Park, Hong Kong, China; 5Shenzhen Research Institute, The Hong Kong Polytechnic University, Shenzhen 518052, China

**Keywords:** neurodegeneration, ganglion cell, optic nerve, glaucoma, proteomics, mass spectrometry

## Abstract

To explore the temporal profile of retinal proteomes specific to primary and secondary retinal ganglion cell (RGC) loss. Unilateral partial optic nerve transection (pONT) was performed on the temporal side of the rat optic nerve. Temporal and nasal retinal samples were collected at 1, 4 and 8 weeks after pONT (n = 4 each) for non-biased profiling with a high-resolution hybrid quadrupole time-of-flight mass spectrometry running on label-free SWATH^TM^ acquisition (SCIEX). An information-dependent acquisition ion library was generated using ProteinPilot 5.0 and OneOmics cloud bioinformatics. Combined proteome analysis detected 2531 proteins with a false discovery rate of <1%. Compared to the nasal retina, 10, 25 and 61 significantly regulated proteins were found in the temporal retina at 1, 4, and 8 weeks, respectively (*p* < 0.05, FC ≥ 1.4 or ≤0.7). Eight proteins (ALDH1A1, TRY10, GFAP, HBB-B1, ALB, CDC42, SNCG, NEFL) were differentially expressed for at least two time points. The expressions of ALDH1A1 and SNCG at nerve fibers were decreased along with axonal loss. Increased ALDH1A1 localization in the inner nuclear layer suggested stress response. Increased GFAP expression demonstrated regional reactivity of astrocytes and Muller cells. Meta-analysis of gene ontology showed a pronounced difference in endopeptidase and peptidase inhibitor activity. Temporal proteomic profiling demonstrates established and novel protein targets associated with RGC damage.

## 1. Introduction

Once the central nervous system (CNS) is injured, the damaged neurons die through primary degeneration, the glial cells are rapidly activated. and the physiological conditions of the neural tissue are adversely altered [1,2,3]. The preceding chain reactions and responses in adjacent anatomical regions lead to the spread of signaling and, subsequently, the secondary degeneration of non-injured neurons [4,5,6,7,8,9]. In glaucomatous optic neuropathy, retinal ganglion cell (RGC) axons at a localized area of the optic nerve are impacted frequently by elevated intraocular pressure (IOP) and may decide to degenerate [10,11]. The initial loss of RGCs results in a deficit of visual functions in patients. Unfortunately, if left untreated, secondary RGC degeneration follows and leads to widespread loss of RGCs over a period of time [12,13,14,15]. In order to develop an effective treatment to prevent blindness, a better understanding of the mechanisms of primary and secondary RGC degeneration is definitely needed.

It Is a challenging task to study primary and secondary degeneration within neural tissue. Multiple laboratories utilized a rat model of partial optic nerve transection (pONT), which was established to morphologically distinguish secondary degeneration from primary RGC degeneration [16,17,18]. In the area of primary RGC degeneration, around 1 to 2 weeks, demyelination, gliosis, and axonal loss is noticed in the optic nerve, while a dramatic loss of RGC bodies is detected in the retina. In the following weeks, the delayed loss of RGC bodies is found at a location beyond the initial injury site and this adjacent area is considered secondary RGC degeneration [19,20]. Microarray analysis has revealed molecular differences in the responses between these two processes, including innate inflammatory cell recruitment and infiltration, the activation of Casp3, both intrinsic and extrinsic apoptosis-associated gene expression, and antioxidant activity [21,22]. Our recent investigation demonstrated that primary degeneration induced acute RGC loss (21.1%) at 1 week, and the total loss increased to 72.3% at 8 weeks, while no change in the number of RGC bodies was detected in the intact area at 1 week but delayed loss (43.6%) was found at 8 weeks for secondary degeneration [23]. To explore the molecular difference between the early phase of the two processes, our laboratory employed 2D Fluorescence Difference Gel Electrophoresis (DIGE) and discovered that eight proteins were significantly regulated at secondary degeneration at 2 weeks, whereas the expressions of three proteins were altered at primary degeneration [23]. The majority of significantly regulated proteins were stress proteins, suggesting that the regional induction of crystallins function to protect neurons against the noxious effects from the primary injury site. Our recent investigation used an integrated approach with label-free sequential window acquisition of all theoretical mass spectra (SWATH-MS) for unbiased proteomic analysis and identified more proteins regulated in primary RGC degeneration (63 proteins) than secondary RGC degeneration (25 proteins) at the same time point [24]. The identities of regulated proteins were confirmed via targeted multiple reaction monitoring (MRM)-MS and the results showed that only six regulated proteins were shared by these two processes. The predicted pathways in primary RGC degeneration (for example, ferroptosis, the HIF-signaling pathway, the adipocytokine signaling pathway, necroptosis, and cell adhesion molecules) are substantially different from secondary RGC degeneration pathway (for example, complement and coagulation cascades, hepatitis B, the Jak-STAT signaling pathway, pancreatic cancer, and serotonergic synapse), supporting the notion that the two degeneration processes are diverse in nature. However, the detailed information regarding how the retinal protein expressions are altered in regions exposed to direct and indirect injury is still lacking. Therefore, the present study aimed to examine the temporal change in protein regulation corresponding to primary and secondary RGC degeneration and characterize their spatial and cellular contribution.

## 2. Results

Topographical and temporal loss of RGC somas in the retinas and axons in the optic nerves was previously documented [23]. Primary and secondary RGC degeneration after pONT has been distinguished using histological methods. For primary RGC degeneration, more rapid loss of RGC somas and axons was detected at the region of initial injury, which was located at the temporal side. The delayed loss of RGCs was observed at the nasal side and considered secondary RGC degeneration. Compared to controls, their molecular phenotypes at 2 weeks after pONT were remarkably different as examined by two different proteomic approaches, namely quantitative 2DIGE [23] and label-free combined method of SWATH-MS and MRM-MS [24]. To understand the propagation of RGC loss, the present experiment compared the retinal protein regulation in primary and secondary RGC degeneration at various time points after pONT procedures.

After individual retinal quadrants were harvested and processed independently for protein extraction, the workflow for protein identification and quantification by applying an untargeted approach using a nanoLC Triple TOF 6600 LCMS was followed, as shown in Figure 1. The numbers of proteins and peptides for individual search analysis at 1 week, 4 weeks and 8 weeks identified using label-free SWATH-MS are summarized in Table 1. Differentially regulated protein was determined by comparing the protein expression in the temporal quadrant to the nasal quadrant based on our statistical criteria. The number of differentially expressed proteins at 1 week, 4 weeks and 8 weeks after pONT was 10, 25 and 61, respectively. Overall, 2531 proteins (18,871 peptides) were identified when all samples were pooled for combined search analysis among three time points.

The identities of significantly regulated proteins at 1 week, 4 weeks, and 8 weeks are listed in Table 2, Table 3 and Table 4, respectively. The ranges of fold changes of regulated proteins for 1 week, 4 weeks, and 8 weeks were 0.11–9.12, 0.08–17.25, and 0.06–13.76, respectively. For all three time points, more upregulated proteins were detected at the temporal retinal quadrant than downregulated proteins. The fold change profile of retinal proteins quantified in samples for 1 week, 4 weeks, and 8 weeks are shown in the volcano plots in Figure 2A–C, respectively. Consistent with our previous report, significant differentiation with a stringent cutoff line at a 1% false discovery rate (FDR) with *p* < 0.05 and the value of Log2 fold change (FC) ≥ 0.43 or ≤−0.43, with at least two quantifiable peptides per protein, was used for analysis.

Retinal proteins that were significantly regulated for at least two time points were further analyzed by immunohistochemistry. Venn diagram (Figure 3) summarized that one protein (ALDH1A1) was upregulated at 1 week and 4 weeks while seven proteins (TRY10, GFAP, HBB-B1, ALB, CDC42, SNCG, and NF-L) were significantly changed at 4 weeks and 8 weeks. As expected, more protein regulations were found with longer treatment times. No protein was differentially regulated for more than two time points with our stringent criteria.

Figure 4 shows the fold change profiles of eight proteins that exhibited differential regulation across all three time points. These highly confident proteins fulfilled the set significance criteria at least for two time points. Five out of eight proteins were upregulated (ALDH1A1, TRY10, GFAP, HBB-B1, ALB), while three proteins were downregulated (CDC42, SNCG, NF-L) within an 8-week period.

For meta-analysis of Gene Ontology (GO), Table 5 lists the top 10 categories of biological processes, molecular function, and cellular components with significant changes at 8 weeks after pONT. The *p*-values of these categories for 1 week and 4 weeks after pONT were also included. For 4 weeks, the genes within four categories under GO term-biological process were significantly regulated (response to wounding, visual system development, sensory system development, eye development), while three categories under GO term-molecular function (serine-type endopeptidase inhibitor activity, endopeptidase inhibitor activity, peptidase inhibitor activity) and cellular component (extracellular space, extracellular region, intermediate filament) were differentially expressed. For 1 week, the genes within two categories under GO term-biological process (negative regulation of endopeptidase activity, negative regulation of peptidase activity) and four categories under GO term-molecular function (endopeptidase inhibitor activity, peptidase inhibitor activity, endopeptidase regulator activity, peptidase regulator activity) were significantly changed whereas none of GO term-cellular component was statistically significant. There was a strong tendency of an increasing number of differential changes in biological process, molecular function, and cellular component as the degeneration progresses. Interestingly, the genes under the categories of endopeptidase inhibitor activity and peptidase inhibitor activity in GO terms-molecular function were significantly regulated for all three time points.

To visualize the localization of regulated proteins, fluorescence immunohistochemistry was performed. The intensity and location of immunoreactivity in the temporal and nasal retina at times were compared. In the control retina, gamma-synuclein (SNCG) was predominantly expressed in the nerve fibers (Figure 5). At 4 and 8 weeks after pONT, a greater reduction in SNCG immunoreactivity was noted in the temporal retina rather than in the nasal retina. At 8 weeks, double labeling of SNCG and RBPMS, a marker for RGC body, demonstrated that SNCG was mainly expressed by axon bundles but not RGC bodies (Figure 6).

Positive GFAP immunoreactivity was mostly limited to the inner limiting membrane of the control retina. The immunoreactivity of GFAP was increased at times after pONT and appeared more extensive in the temporal retina relative to the nasal retina (Figure 7). At 4 weeks, GFAP immunoreactive vertical processes were noted in the inner plexiform layer (IPL) of both the temporal and nasal retina, but only some vertical processes were detected in the inner nuclear layer (INL) of the temporal retina. At 8 weeks, GFAP immunoreactive vertical processes were found in the IPL and INL layer of both the temporal and nasal retina; however, there was a few GFAP-positive vertical processes only in the outer nuclear layer (ONL) of the temporal retina. The double labeling of GFAP and S100, a marker for Muller cells, was performed on the retina at 8 weeks and the control retina (Figure 8). The co-localization of GFAP and S100 was present at the inner limiting membrane of control retina, and GFAP processes in IPL were S100-negative. However, the vertical processes in the ONL of temporal retina and the INL of the nasal retina were labeled by both GFAP and S100 at 8 weeks after injury.

In the control retinas, ALDH1A1 immunoreactivity was noted in the NFL (Figure 9). Scattered ALDH1A1 immuno-positive cell bodies and processes were observed in the INL and ONL of the temporal retina at 1 week. At 4 and 8 weeks, ALDH1A1 labeling in the NFL was diminished, and there was an increased number of positive labeled cells in INL and processes in ONL. The temporal retina had more ALDH1A1-labeled cell bodies and processes than the nasal retina.

## 3. Discussion

The data-independent acquisition (DIA) approach of quantitative proteomic analysis was used to provide a non-biased comprehensive evaluation in order to understand how the injured RGCs are dying and how the neighboring intact RGCs are affected. As previously confirmed via MRM-MS, the label-free SWATH-MS technique offers high accuracy and consistency in quantitative proteomic analysis of retinal tissues after localized optic nerve injury [24]. In particular, the present study examined the temporal profile of protein expression change in the injured and adjacent non-injured retinal areas and demonstrated the differential cellular responses between primary and secondary RGC degeneration after pONT. In general, there was a strong tendency of increasing differential protein regulation between primary and secondary RGC degeneration process as it progresses. Eight weeks after pONT, eight retinal proteins, including ALDH1A1, TRY10, GFAP, HBB-B1, ALB, CDC42, SNCG, and NF-L, were found to be significantly regulated for at least two time points. The localization of these proteins indicate the involvement of the nerve bundles of RGCs, astrocytes, and Muller cells. Meta-analysis of Gene Ontology showed extensive alterations in molecular function, cellular component, and biological processes at 8 weeks and predicted that the genes associated with endopeptidase inhibitor activity and peptidase inhibitor activity are significantly regulated within the first week up until the eighth week representing a major characteristic to distinguish between primary and secondary RGC degeneration.

The decreased expression of SNCG in the temporal retinal quadrant was observed at 1 week, 4 weeks, and 8 weeks when compared to the nasal retinal quadrant (Figure 4). Statistically, the downregulation of SNCG at 1 week showed a marginal lack of significance to pass the confidence filter (0.63, which is slightly lower than the set criteria at 0.70) and, therefore, differential downregulation was considered at 4 and 8 weeks only in this study. This could be partially due to the relatively weak MS signal of this particular protein. While it is highly abundant in the brain and periphery nervous system, SNCG is a cytoskeleton-associated protein and is responsible for the maintenance of neurofilament integrity and axonal architecture [25]. Specific SNCG expressions in RGCs has been described in mice [26], humans [27], and macaques [28]. The expression of SNCG has believed to be a specific marker for RGCs [27]. A recent study using AAV-mediated CRISPR gene therapy demonstrated that mouse gamma–synuclein promoters directly modulated endogenous degenerative genes to preserve the acutely injured RGC somata and axons [29]. Our experiment using immunohistochemistry showed that SNCG is predominantly expressed by the RGC axons at the retinal nerve fiber layer. Similar to optic nerve axotomy and elevated intraocular pressure-induced injury, the present finding supports that the loss of RGCs is associated with the downregulation of SNCG expression in the retina after optic nerve injuries [30]. The expression level of SNCG has been correlated with the progression of some cancers [31] and Alzheimer’s disease [32], suggesting a role in the pathophysiology. Aggregated SNCG was found in the motor neurons of some amyotrophic lateral sclerosis (ALS) patients [33]. In addition, SNCG immunopositive glial inclusions appeared in the optic nerve of glaucoma patients [34]. Therefore, it is hypothesized that the accumulation of SNCG in other tissue locations may be associated with the interruption of signal transduction pathways, such as Elk-1 and MAPK, or the activation of the extracellular matrix remodeling, such as matrix metalloproteinases [35,36].

Similarly, neurofilament light polypeptide (NF-L) expression was reduced in the temporal retinal quadrant at 4 and 8 weeks after pONT. It is known that NF-L, together with NF m (medium) and NF-H (high), forms part of the neuronal intermediate neurofilament triplet, and the neurofilament triplet assembles in the neuronal soma and travels through slow axonal transport depending on phosphorylation [37]. In the mammalian retina, NFs are preferentially expressed in RGCs, and differential phosphorylation of these structural proteins has been used for labeling RGC somas and their intraretinal axons and for the identification of degenerating RGCs [38,39,40]. Presumably, NF-L expression is localized to the nerve fiber layer and also relates to the loss of RGCs [41]. A phosphorylated NF immunohistochemistry experiment was performed (unpublished data), and the pattern change in phosphorylated NF was similar to SNCG expression after pONT. Clearly, together with the finding of SNCG regulation, the decreased level of structural proteins for axons demonstrates the severity of RGC loss after optic nerve injury.

Glial fibrillary acidic protein (GFAP), an intermediate filament, is mostly expressed by the retinal astrocytes in the inner limiting membrane of the normal retina, as visualized via immunohistochemistry [42]. Marked upregulation of GFAP and increased dendritic complexity of the astrocytes is a common phenomenon in response to many retinal injuries indicating its potential role in glial remodeling [43]. In contrast, Müller cells at the resting stage do normally produce GFAP at very low levels but are hardly detected via the immunohistochemical method [44]. In response to retinal damage or microenvironmental change, rapid upregulation of GFAP in Müller cells is more noticeable [45,46]. In agreement with the increased expression of GFAP, as analyzed using mass spectrometry, an increased number of GFAP-positive Muller cell bodies and increased immunoreactivity of GFAP in radial processes of Muller cells are evident at 4 and 8 weeks after pONT. Compared to the nasal retinal quadrant, the upregulation of GFAP in both glial cell types is more prominent in the temporal quadrant, indicating that GFAP upregulation is not a panretinal effect in response to local stress. The stronger reactivity of Müller cells at the primary RGC degeneration may explain more excessive levels of released neurotransmitter glutamate or less effective detoxification due to greater RGC loss at this region compared to the secondary RGC degeneration region [12].

The remarkable upregulation of ALDH1A1 was noted in the temporal retinal quadrant as early as 1 week and up to 4 weeks after pONT. ALDH1A1 appears to be present in the nerve fiber bundle in the normal retina, but the ALDH1A1 labeled nerve fiber becomes thinner following injury. However, ALDH1A1-labeled cell bodies and processes in INL and ONL are more prominent at 4 and 8 weeks. There is some discrepancy between increased levels of ALDH1A1 as analyzed by MS methodology and staining pattern using immunohistochemical technique. In general, the ALDH superfamily represents a group of enzymes that comprises a wide variety of NAD(P+)-dependent enzymes which could catalyze the oxidation of aldehydes to their corresponding carboxylic acids [47]. It is believed that through their catalytic functions, ALDH enzymes protect cells by detoxifying reactive aldehydes and also modulate embryogenesis and neurotransmission [48]. They function as important components of cellular defense mechanisms against ultraviolet radiation, and reactive oxygen species induced corneal damage and protect retinal cells against lipid peroxidation (LPO) [49]. The mechanisms leading to the upregulation of ALDH1A1 are not completely understood but possibly relate to antioxidative response [50]. ALDH1A1 recognized retinaldehyde substrates and aldehydes derived from LPO, such as 4-hydroxy-2-nonenal (4-HNE) [51]. 4-HNE was shown to modify Keap-1 protein and activate transcription factor Nrf2, the central regulator of antioxidant-responsive elements response [52], which in turn regulates the expression of ALDH [53]. This may explain the upregulation of ALDH1A1 in the retina after pONT. Alternatively, ALDH1A1 regulation is a physiological and pathological target of Cyclin-dependent kinase-5 (CDK5), which is an isoform associated with retinal homeostasis [54]. As shown, CDK5 is maximally expressed in the RGC layer and likely upregulates ALDH1A1 transcription under neurotoxic conditions [55]. On the other hand, ALDH1A1 genes were also found to be expressed principally by Müller glia in diabetic retinopathy [56]. The blockade of ALDH1A1 in cultured Müller glia reduced cell viability by triggering FDP-lysine accumulation. Overall, ALDH1A1 apparently may play a neuroprotective role in both RGCs and Muller cells against noxious stress.

The weakness of the present study is that the patterns of several regulated proteins cannot be characterized using immunohistochemistry due to the limited choice of commercial antibodies and undesirable optimization resulting from heavy background staining and cross-reactivity. Although our present study did not provide immunohistochemistry finding for albumin, Beta-glo, RCG64250, and cell division control protein 42 homolog, the data of these protein expressions revealed using SWATH-MS strongly suggest that their potential roles in primary and secondary RGC degeneration have to be further clarified.

Meta-analysis demonstrated numerous outstanding mechanism differences between primary and secondary RGC degeneration, particularly at 8 weeks after pONT. The top 10 categories under GO term-biological process, molecular function, and cellular component were present in this study. Under the GO term-biological process category, activities related to the development of lens, visual and sensory systems, and negative regulation of endopeptidase, hydrolase and peptidase activity are altered at 8 weeks. Under the GO term molecular function category, activities related to endopeptidase and peptidase inhibitors were involved not only at 8 weeks but also 1 week and 4 weeks. Neuroserpin is an axonally secreted serpin that is involved in regulating plasminogen and its enzyme activators, such as tissue plasminogen activator (tPA) [57]. The protein has been increasingly shown to play key roles in neuronal development, plasticity, maturation, and synaptic refinement. The proteinase inhibitor may function both independently and through tPA-dependent mechanisms. It is interesting to note that expressions of Serpina1, Serpina3k, Serpina4, and Serpinc1 were found to be regulated at 8 weeks, while there was Serpina3l regulation at 4 weeks and Serpinb6a and Serpina3l regulation at 1 week. It is tempting to propose that these genes may serve as early biomarkers, and their differential regulation may reflect the severity stage of the disease. Recently, neuroserpin inclusion bodies (FENIBs) have been linked to Alzheimer’s disease, cancer, glaucoma, stroke, neuropsychiatric disorders, and familial encephalopathy [57,58,59]. Understanding the detrimental consequences of neuroserpin dysfunction and alterations and cellular signaling networks is also essential to developing new therapy for slowing the progression of RGC loss. In addition, the findings from the cellular component category demonstrated significant differences in activities associated with hemoglobin, intermediate filaments, and neurofilaments at 8 weeks, confirming the cellular contribution from the neurons and glial cells in the progression of RGC loss.

To sum up, non-biased proteomic analysis revealed quantitative changes in protein levels in dynamic and progressive processes specific to primary and secondary RGC degeneration after localized optic nerve injury. The change in protein expression patterns demonstrated the involvement of multiple retinal cell types, including neurons and glial cells. The protein library database of this experimental model contributes to the understanding of the mechanisms responsible for acute and delayed neurodegeneration and will be useful for drug studies targeting the pathways specific to primary and secondary RGC degeneration.

## 4. Materials and Methods

### 4.1. Animals and Partial Optic Nerve Transection (pONT)

All animal procedures were performed in accordance with the ARVO statement for the Use of Animals in Ophthalmic and Vision Research and the policies of the UCLA Animal Research Committee. Adult Wistar rats (3 months-old; 300–350 g) were housed with standard food and water provided ad libitum in the animal research facility of the University of California Los Angeles and kept for at least one week prior to procedures.

After anesthesia with isoflurane gas and topical 1% proparacaine eye drops, an incision was made in the temporal conjunctiva for access to the retrobulbar optic nerve, using a diamond knife to incise the optic nerve to a depth of one-third of its diameter 2–3 mm behind the globe, as described previously [23,24]. Following conjunctival suturing, an ophthalmoscopic examination was performed to ensure complete retinal blood flow. Prophylactically, topical tobramycin (Alcon, Fort Worth, TX, USA) was applied immediately and then twice daily for 2 days. Animals were sacrificed via carbon dioxide overdose with cervical dislocation as the secondary euthanasia method to ensure complete euthanasia. For each animal, surgical procedures were performed on one eye, whereas the contralateral eye was untreated. Both eyeballs were enucleated for analyses.

### 4.2. Rat Retinal Harvest and Protein Extraction

Twelve animals were divided into three groups (n = 4 per group) in 1 week, 4 weeks, and 8 weeks groups, respectively. Dissected retinas were divided into four retinal quadrants regions (Superior, Inferior, Nasal, and Temporal). Aiming to investigate the difference between primary and secondary RGC degeneration after pONT, the present experiment was carried out to compare the molecular and morphological changes in the temporal (T) and nasal (N) retinal quadrants only. Four untreated control rat retinas were also included for quality control and dissected accordingly. Retinal tissues were homogenized with 100 µL lysis buffer (7 M urea, 2 M thiourea, 2% CHAPS, 30 mM Tris, 0.2% Biolytes, 1% dithiothreitol, 1% ASB14, and protease inhibitor cocktail) using Precellys 24 homogenizer (Bertin Technologies, Aix-en-Provence, France) at 5800 rpm twice for 30 s with an interval of 20 s. The homogenized samples were then centrifuged at 218,000× *g* for 25 min at 4 °C. Only the supernatant was collected for further analysis. The protein concentration of each sample was measured using the Bradford assay (Bio-Rad, Hercules, CA, USA).

### 4.3. Protein Reduction, Alkylation, and Digestion

Thirty-five micrograms (35 µg) of protein from each retina sample was reduced in 8 mM dithiothreitol (DTT) in 25 mM NH4HCO3 for 45min at 37 °C, followed by alkylation in 20 mM iodoacetamide (IAA) in 25 mM NH4HCO3 for 30 min at room temperature in the dark. The samples were then cleaned up by adding 4× volume of ice-cold acetone at −20 degrees overnight. After centrifugation at 218,000× *g* for 25 min at 4 °C, the supernatant was removed. The protein pellet was washed once with 80% acetone and re-dissolved in 1 M urea with 25 mM NH_4_HCO_3_. Proteins in retinal samples were in-solution digested with trypsin with a final ratio of 1:10 (trypsin: protein amount, *w*/*w*) at 37 °C for 16–18 h. Four biological samples from the same group and region were mixed to form a representative pooled sample. At last, six representative samples of three groups (one representative sample of the temporal and nasal retinal quadrant in each group) from the experimental eyes (1 week, 4 weeks, and 8 weeks) were generated. Peptides were purified using an Oasis HLB cartridge (Waters Associates, Inc., Milford, MA, USA) and re-suspended in 20 µL 0.1% formic acid (FA). The total peptide assay was measured using Pierce Quantitative Colorimetric Peptide Assay Kit (Thermo Scientific, Rockford, IL, USA) according to the manufacturer’s instructions. The final peptide concentration of each sample was calibrated to 0.5 µg/µL before injection into mass spectrometry (MS).

### 4.4. Data Dependent Acquisition (DDA) and SWATH-MS Acquisition

Both information-dependent acquisition (IDA) and SWATH-MS acquisitions were performed using a TripleTOF 6600 system (Sciex, Framingham, MA, USA) with Analyst TF (Sciex, Framingham, MA, USA, version 1.7) fitted with a Nanospray III source, which was equipped with a hybrid quadrupole time-of-flight mass analyzer as previously described [60]. The peptide sample was loaded onto a trap column (350 µm × 0.5 mm, C18) by loading the buffer (0.1% FA, 2% acetonitrile in water) at a speed of 2 µL/min for 15 min. It was then separated on a nano-LC analytical column (100 µm × 15 cm, C18, 5 µm) using an Ekisgent ekspertTM 415 nano-LC system. LC separation was performed under 350 nL/min using mobile phase A (0.1% FA, 2% ACN in water) and B (0.1% FA, 98% ACN in water) with the following gradient: 0–0.5 min: 5%B, 0.5–90 min: 10%B, 90–120 min: 20%B, 120–130 min: 28%B, 130–135 min: 45%B, 135–141 min: 80%B, 141–155 min: 5%B. The following parameters were applied for DDA: mass scan at 350–1800 m/z with 250 ms accumulation time, MS/MS scan at 100–1800 m/z with 50 ms accumulation time, and charge state between 2 and 4 in high-sensitivity mode. The isolation of fixed 25Da windows was selected in the SWATH acquisition.

For protein identification, a total of 2 µg peptides from six representative samples were injected into MS for the IDA experiment. Data from each IDA file were combined for searching to generate a combined ion library (.group), which was thoroughly searched against the Rattus norvegicus Uniprot database in ProteinPilot^TM^ software (SCIEX, Framingham, MA, USA, version 5.0) utilizing the Paragon algorithms with the following parameters: trypsin as the digested enzyme, alkylation using iodoacetamide, and 1% global false discovery rate (FDR). The resulting group file was set as the ion library file for SWATH data quantification.

### 4.5. SWATH-MS and Statistical Analysis

For protein quantification, 2 µg tryptic digested peptides in each sample were used for SWATH-MS acquisitions with two technical replicates. An IDA was imported as the reference ion library. Variable isolation of 100 windows was applied in a full mass range of 100–1800 m/z scan in SWATH acquisition. All raw SWATH data of 1, 4, and 8 weeks (temporal quadrants compared to their nasal sides) were uploaded to the OneOmics™ Suite platform (SCIEX, Framingham, MA, USA, version 3.4) via CloudConnect for quantitative analysis. Proteins were considered as differentially expressed proteins if they met these three criteria: fold change (log_2_FC ≥ 0.43 or ≤−0.43), confidence ≥0.70, and *p*-value < 0.05, using a *t*-test. The cut-off threshold of FC and *p*-value were the same as our previously published protocol [24].

### 4.6. Bioinformatics Analysis

All of the identified proteins upon pONT treatment were converted to corresponding gene names using the batch id conversation tool in the Universal Protein Resource online database (UniProt, http://www.uniprot.org/, last modified on 14 December 2022) before bioinformatics analysis. To compare and contrast changes of proteins involved in RGC degeneration after pONT treatment for 1 wk, 4 wk, and 8 wk, the data set of all regulated proteins was further analyzed using meta-analysis in iPathwayGuide (http://www.advaitabio.com/, accessed on 15 February 2023). Gene Ontology (GO) annotations of quantified proteins were also classified using iPathwayGuide software (Advaita Corporation, Ann Arbor, MI, USA), which can identify unique/common traits quickly across three time points. A threshold of *p*-value < 0.05 and log_2_FC ≥ 0.43 or ≤−0.43 were employed.

### 4.7. Tissue Processing and Immunohistochemistry

To visualize the change in retinal protein localization after pONT, twenty-four animals were used and divided into 4 groups including 1 week, 4 weeks, and 8 weeks after pONT, and untreated control (n = 6 per group). After euthanasia, the eyeballs were carefully enucleated, and the suture at the temporal conjunctiva remained, showing orientation. The eyeballs were fixed with 4% paraformaldehyde in 0.1 M phosphate buffer, cryoprotected in 30% sucrose and embedded. Serial ten-µm thick retinal sections with optic nerve head at the horizontal plane were cut with cryostat and collected. Each retinal section consisted of temporal and nasal retinal quadrants, which corresponded to primary and secondary RGC degeneration, respectively.

For immunohistochemical procedures, the samples were incubated with 10% fetal bovine serum in 0.1% Triton X-100 in PBS for 30 min to block non-specific staining and then with primary antibodies at 4 °C overnight, as previously described [61]. Commercially available antibodies were purchased and optimized. The primary antibodies used were RNA-binding protein with multiple splicing (RBPMS; 1:500; rabbit; ProSci, Polway, CA, USA), glial fibrillary acidic protein (GFAP; 1:1000; chicken; Abcam, Waltham, MA, USA), gamma-synuclein (SNCG; 1:500; rabbit; Abcam, Waltham, MA, USA), aldehyde dehydrogenase 1A1 (ALDH1A1; 1:500; rabbit; Sigma, St Louis, MO, USA), and S100-beta (S100; 1:500; rabbit; Abcam, Waltham, MA, USA) were used. After incubation with the primary antibodies, the sections were washed with 0.1% Triton X-100 in PBS and incubated with secondary antibodies for 1 h at room temperature. Secondary antibodies used in this study: Alexa Fluor 488-conjugated goat anti-rabbit IgG, Alexa Fluor 568-conjugated goat anti-mouse IgG and Alexa Fluor 488-conjugated goat-anti-guinea pig IgG (1:500; Thermo Fisher Scientific, Canoga Park, CA, USA). The sections were mounted with a medium containing DAPI for nuclear counterstaining and imaged with a fluorescence microscope (Revolve, ECHO, San Diego, CA, USA). The pattern and intensity of positive immunoreactivity were evaluated.

## Figures and Tables

**Figure 1 ijms-24-12109-f001:**
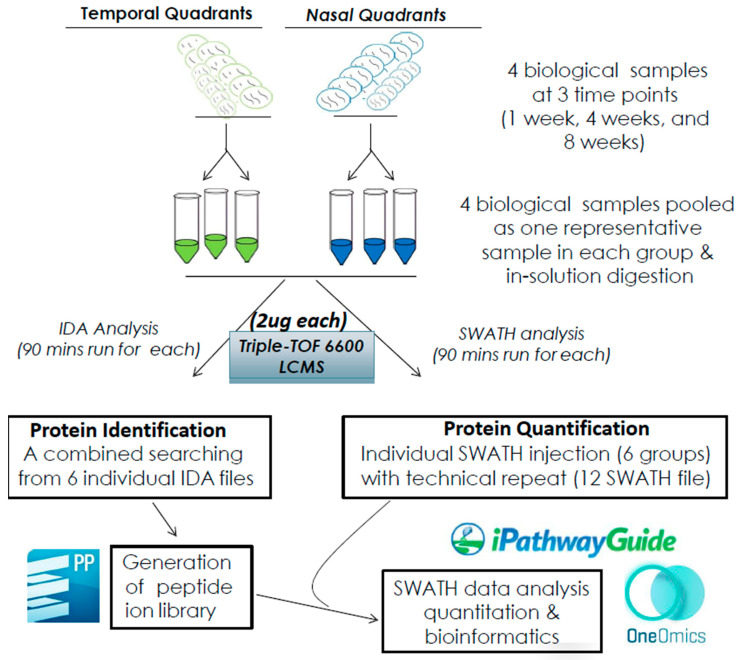
Workflow diagram for SWATH and bioinformatics analysis. Four biological samples of temporal and nasal quadrants at three time points (1 wk, 4 wk, and 8 wk) were pooled to form six representative samples. Technical duplicates were included. Two micrograms of digested peptides were identified via ProteinPilot (PP) and quantified via the OneOmics Cloud platform. Subsequent bioinformatics analysis was performed using iPathwayGuide.

**Figure 2 ijms-24-12109-f002:**
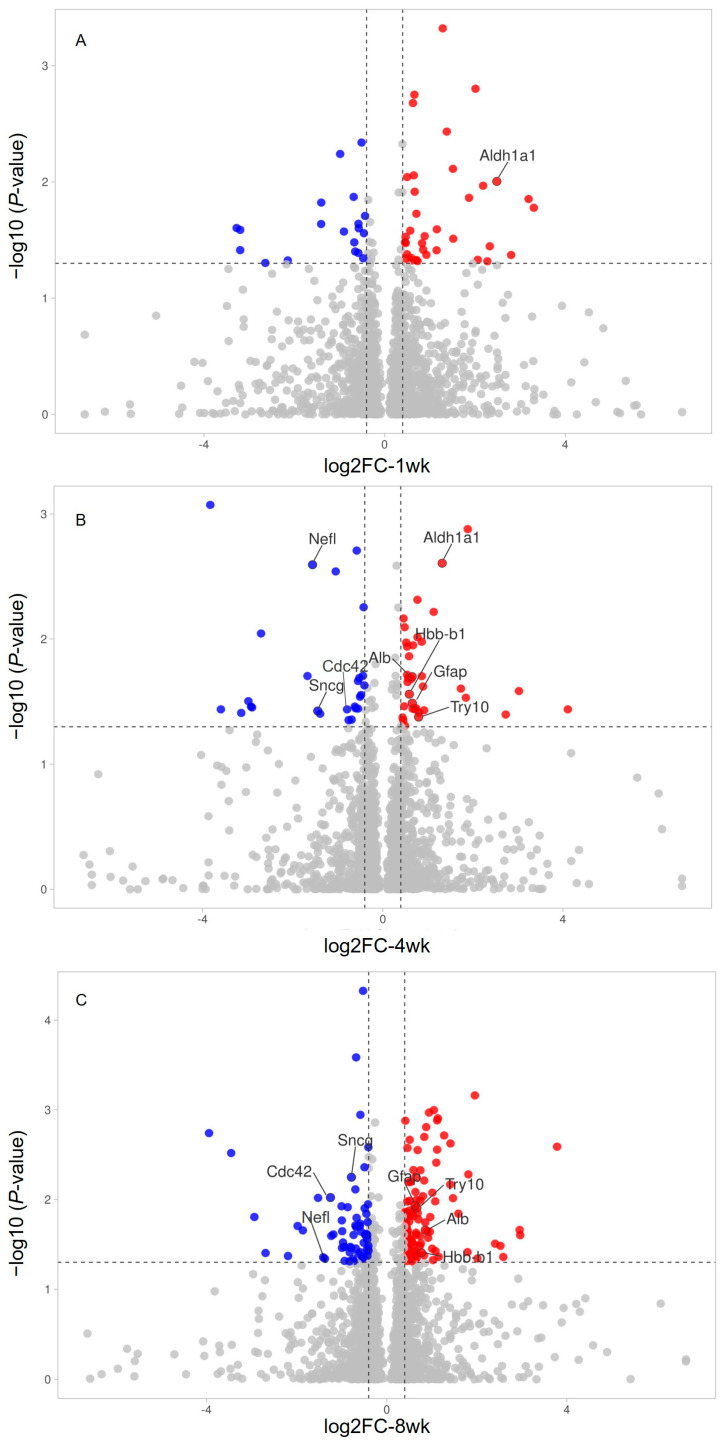
Volcano plot of retinal proteins quantified using SWATH-MS (shown in gene names) in samples at 1 wk (**A**), 4 wk (**B**), and 8 wk (**C**) after pONT. The horizontal axis is the log_2_(Fold Change), and the vertical axis is the negative log10 value of the *p*-value. The dashed lines represent the threshold (0.43 and −0.43 for the *x*-axis and 1.30 for the *y*-axis). The upregulated proteins (log_2_FC ≥ 0.43 and *p* < 0.05) are shown in red, while the downregulated proteins (log_2_FC ≤ −0.43 and *p* < 0.05) are in blue. Names of regulated proteins shared by 2 time points are shown. Fold changes in all groups are calculated by comparing temporal compared to nasal quadrants (T/N).

**Figure 3 ijms-24-12109-f003:**
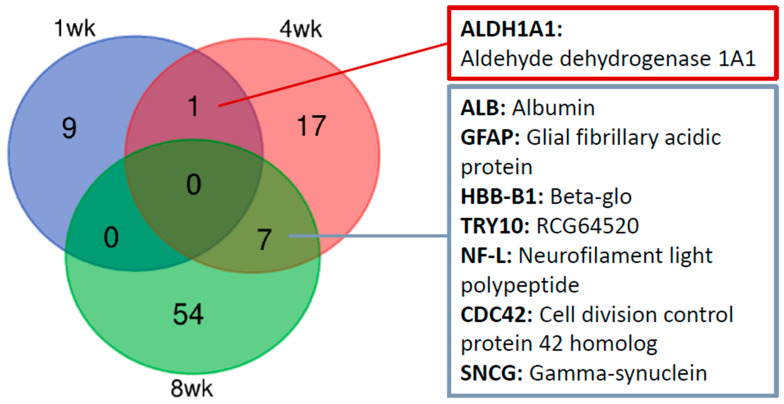
Comparison of differentially expressed proteins among three time points (1 wk, 4 wk, and 8 wk) after pONT. Venn diagram showed that one common protein was shared by 1 wk and 4 wk while seven common proteins were shared by 4 wk and 8 wk. The different colors indicate different time points: 1-week (blue); 4-week (red); 8-week (green). Significantly expressed proteins were considered if the following criteria were met: *p*-value < 0.05, log_2_FC ≥ 0.43 or ≤−0.43, and confidence ≥ 0.70.

**Figure 4 ijms-24-12109-f004:**
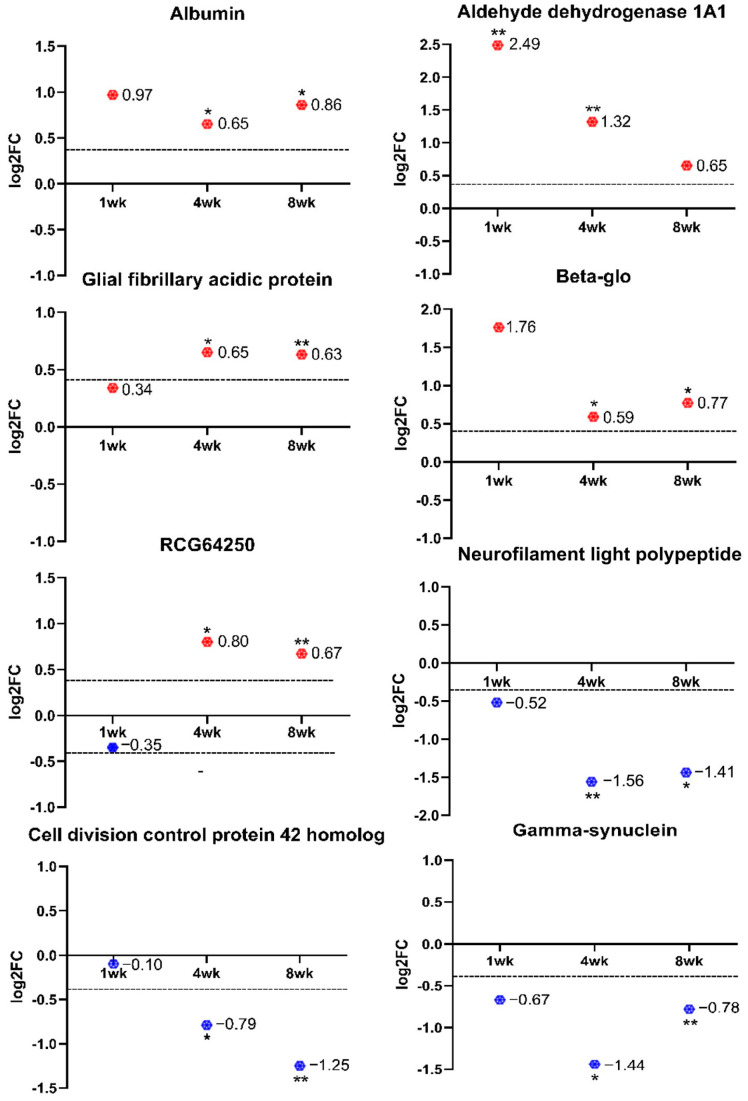
Expression levels of eight proteins that were differentially expressed when compared temporal retinal quadrant to nasal retinal quadrant using the SWATH-MS approach for at least 2 time points among 1 wk, 4 wk, and 8 wk after pONT. The proteins with log_2_FC > 0 are shown in red, while the proteins with log_2_FC < 0 are in blue. The dashed lines represented the *y*-axis as 0.43 or −0.43. * *p* < 0.05, ** *p* < 0.01.

**Figure 5 ijms-24-12109-f005:**
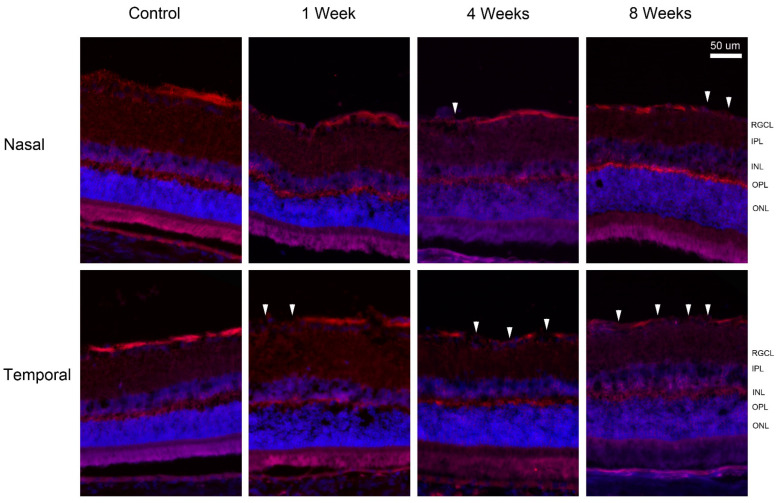
Immunohistochemistry of gamma-synuclein (SNCG) after pONT. Loss of gamma-synuclein immuno-positive nerve fiber (arrowhead) was noted in the temporal retina as early as 1 week up to 8 weeks, while mild loss was observed in the nasal retina at 4 weeks. After pONT, the remaining fibers (red) appeared to be thinner compared to the control. Red = gamma-synuclein; blue = DAPI. RGCL = retinal ganglion cell layer; IPL = inner plexiform layer; INL = inner nuclear layer; OPL = outer plexiform layer; ONL = outer nuclear layer.

**Figure 6 ijms-24-12109-f006:**
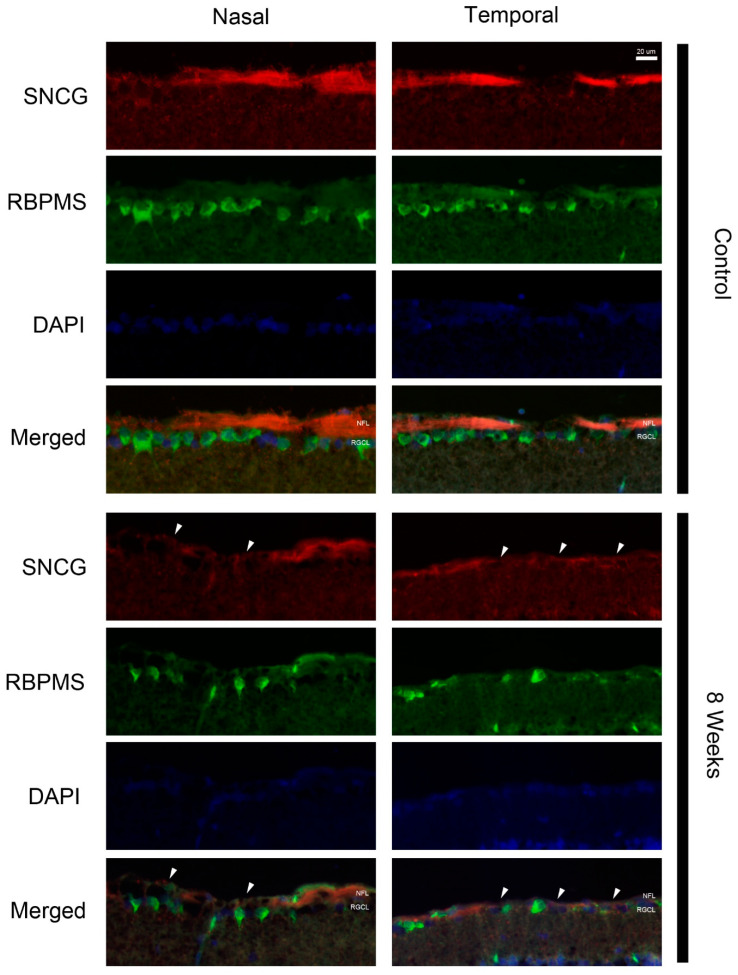
Localization of gamma-synuclein (SNCG) at 8 weeks after pONT. In control retina, gamma–synuclein was expressed by the nerve fibers (red) but not RBPMS-positive RGC bodies (green). After pONT, loss of gamma-synuclein immunolabeling (arrowheads) and dropout of RBPMS-positive cells was more apparent in the temporal retina. Red = gamma–synuclein; green = RBPMS; blue = DAPI. NFL = nerve fiber layer; RGCL = retinal ganglion cell layer.

**Figure 7 ijms-24-12109-f007:**
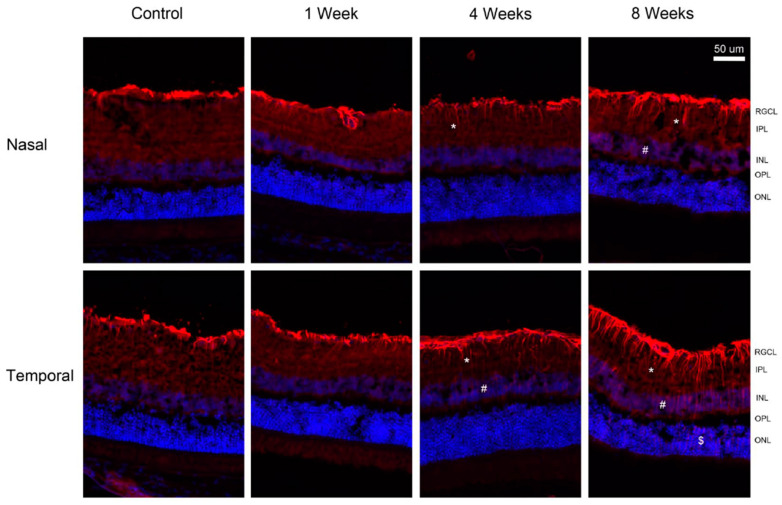
Immunohistochemistry of glial fibrillary acidic protein (GFAP) after pONT. Expression of GFAP was noted in the inner limiting membrane of the control retina. GFAP expression levels were progressively increased, and the processes extended to the inner plexiform layer (*), inner nuclear layer (#) and outer nuclear layer ($) in the temporal retina at 8 weeks. GFAP labeled processes extended up to INL of the nasal retina at 8 weeks. Red = GFAP; Blue = DAPI. RGCL = retinal ganglion cell layer; IPL = inner plexiform layer; INL = inner nuclear layer; OPL = outer plexiform layer; ONL = outer nuclear layer.

**Figure 8 ijms-24-12109-f008:**
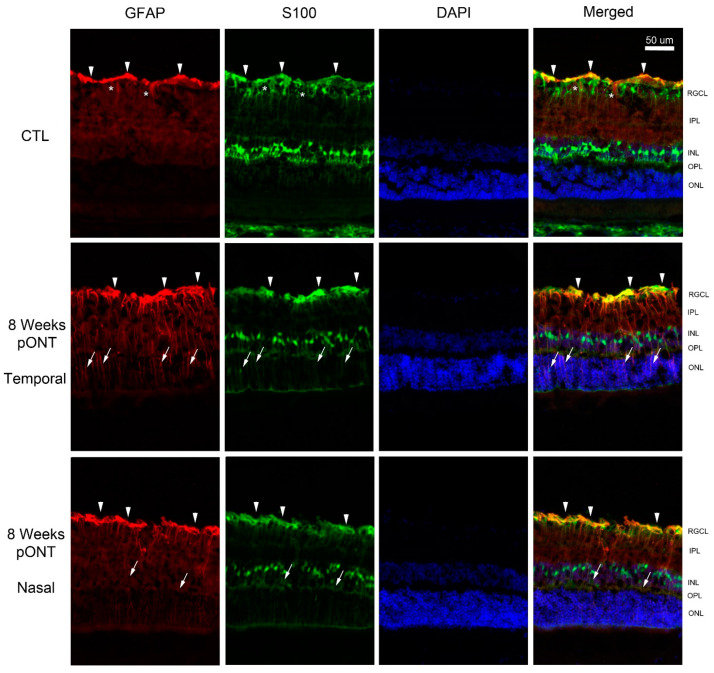
Double labeling of GFAP and S100 at 8 weeks after pONT. Colocalization of GFAP and S100 was found in the inner limiting membrane of the control retina (arrowheads) but some processes were not labeled by S100 (*). At 8 weeks after pONT, there was colocalization of GFAP and S100 at Muller cell processes in the ONL of temporal retinal quadrant (arrows) and a few processes in the INL of nasal quadrant (arrows). Red = GFAP; Green = S100; Blue = DAPI. RGCL = retinal ganglion cell layer; IPL = inner plexiform layer; INL = inner nuclear layer; OPL = outer plexiform layer; ONL = outer nuclear layer.

**Figure 9 ijms-24-12109-f009:**
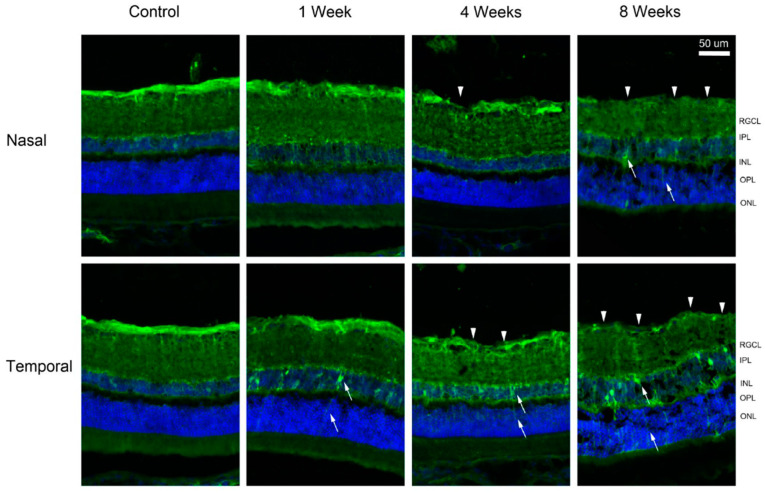
Change in ALDH1A1 expression pattern after pONT. Decreased immunoreactivity was noted in NFL (arrowheads) of both temporal and nasal retina after pONT. More ALDH1A1 labeled cell bodies in INL and processes in ONL (arrows) were found in the temporal retina. Green = ALDH1A1; Blue = DAPI. RGCL = retinal ganglion cell layer; IPL = inner plexiform layer; INL = inner nuclear layer; OPL = outer plexiform layer; ONL = outer nuclear layer.

**Table 1 ijms-24-12109-t001:** Numbers of identified proteins and peptides in 3 individual groups and using a combined search method. Differentially expressed proteins in 3 time points by comparing temporal to nasal quadrants were shown (Temporal/Nasal).

Groups	Protein	Peptide	Differentially Expressed Proteins **
Individual search-1 wk	2023	12,543	10/2208 (8 upregulated and 2 downregulated)
Individual search-4 wk	2173	13,715	25/2208 (16 upregulated and 9 downregulated)
Individual search-8 wk	2185	13,951	61/2208 (47 upregulated and 14 downregulated)
Combined search (1, 4, and 8 wk) *	2531	18,871	N/A

* Number of proteins and peptides using a combined ion library generated from individual samples of 1 wk, 4 wk, and 8 wk. ** Criteria for significant quantification: log_2_FC ≥ 0.43 or ≤−0.43; Confidence ≥ 0.70; *p*-value: <0.05; with at least 2 peptides per protein.

**Table 2 ijms-24-12109-t002:** List of significantly regulated proteins in the temporal quadrant compared to the nasal quadrant after pONT for 1 week. A total of 8 proteins were upregulated (log_2_FC ≥ 0.43) and 2 proteins were downregulated (log_2_FC ≤ −0.43). FC: fold change, T/N: Temporal/Nasal.

Uniprot ID	Gene Name	Protein Name	FC (T/N)	log_2_FC	Confidence	*p*
Upregulation						
D4A8G0	*Lsm12*	LSM12 homolog	9.12	3.19	0.84	0.01
Q1W1V7	*N/A*	Liver cancer-related protein	6.33	2.66	0.88	0.01
A0A0H2UHP1	*Aldh1a1*	Aldehyde dehydrogenase 1A1	5.60	2.49	0.84	0.01
B2GV57	*Cars2*	cysteine-tRNA ligase	5.03	2.33	0.80	0.04
A0A8I5ZSA8	*ENSRNOG00000069233*	Histone H2B	3.34	1.74	0.91	0.03
P69897	*Tubb5*	Tubulin beta-5 chain	2.60	1.38	0.82	0.00
Q6PED0	*Rps27a*	Ubiquitin-40S ribosomal protein S27a	1.78	0.83	0.93	0.03
A0A0G2K2S2	*Slc2a1*	Solute carrier family 2, facilitated glucose transporter member 1member 1	1.48	0.57	0.77	0.03
Downregulation						
A0A0H2UHU0	*Rps25*	40S ribosomal protein S25	0.38	−1.41	0.72	0.02
Q91XU7	*Cdhr1*	Cadherin-related family member 1	0.11	−3.20	0.79	0.03

**Table 3 ijms-24-12109-t003:** List of significantly regulated proteins in the temporal retinal quadrant compared to the nasal retinal quadrant after pONT for 4 weeks. A total of 16 proteins were upregulated (log_2_FC ≥ 0.43) and 9 proteins were downregulated (log_2_FC ≤ −0.43). FC: fold change; T/N: Temporal/Nasal.

Uniprot ID	Gene Name	Protein Name	FC (T/N)	log_2_FC	Confidence	*p*
Upregulation						
B5DEZ3	*Ccdc126*	Ccdc126 protein	17.25	4.11	0.71	0.04
Q91ZE7	*N/A*	Anion exchange protein	8.58	3.10	0.70	0.00
Q5M842	*IgG-2a*	IgG-2a protein	8.13	3.02	0.70	0.03
Q05175	*Basp1*	Brain acid-soluble protein 1	3.36	1.75	0.80	0.04
A0A0H2UHP1	*Aldh1a1*	Aldehyde dehydrogenase 1A1	2.49	1.32	0.89	0.00
P18437	*Hmgn2*	Non-histone chromosomal protein HMG-17	2.19	1.13	0.72	0.01
Q499T3	*Sirpa*	Sirpa protein	1.89	0.92	0.72	0.04
W4VSR7	*Try10*	RCG64520	1.74	0.80	0.85	0.04
A0A0G2K2V6	*Krt10*	Keratin, type I cytoskeletal 10	1.71	0.77	0.75	0.01
A0A0G2JSY7	*Rho*	Rhodopsin	1.70	0.77	0.84	0.00
A0A0H2UHW4	*Pcnp*	PEST proteolytic signal-containing nuclear protein	1.66	0.73	0.70	0.04
D4A3K5	*H1-1*	Histone H1.1	1.59	0.67	0.88	0.01
P47819	*Gfap*	Glial fibrillary acidic protein	1.57	0.65	0.87	0.03
A0A0G2JSH5	*Alb*	Albumin	1.57	0.65	0.82	0.02
Q6PDU6	*Hbb-b1*	Beta-glo	1.50	0.59	0.82	0.03
K7S2S2	*Hist2h2aa3*	Histone H2A	1.43	0.52	0.79	0.01
Downregulation						
Q6P3E1	*Rps16*	Rps16 protein	0.74	−0.43	0.70	0.03
D3ZK97	*H3f3c*	Histone H3	0.74	−0.43	0.99	0.01
P47728	*Calb2*	Calretinin	0.67	−0.58	0.75	0.00
Q8CFN2	*Cdc42*	Cell division control protein 42 homolog	0.58	−0.79	0.75	0.04
A0A0G2K0T6	*Sncg*	Gamma-synuclein	0.37	−1.44	0.87	0.04
P19527	*Nefl*	Neurofilament light polypeptide	0.34	−1.56	0.81	0.00
Q5XI38	*Lcp1*	Lymphocyte cytosolic protein 1	0.13	−2.92	0.81	0.03
Q3B7U9	*Fkbp8*	Peptidyl-prolyl cis-trans isomerase FKBP8	0.13	−2.98	0.70	0.03
G3V9E4	*Apeh*	Acylamino-acid-releasing enzyme	0.08	−3.59	0.71	0.04

**Table 4 ijms-24-12109-t004:** List of significantly regulated proteins in the temporal retinal quadrant compared to the nasal retinal quadrant after pONT for 8 weeks. A total of 47 proteins were upregulated (log_2_FC ≥ 0.43) and 14 proteins were downregulated (log_2_FC ≤ −0.43). FC: fold change; T/N: Temporal/Nasal.

Uniprot ID	Gene Name	Protein Name	FC (T/N)	log_2_FC	Confidence	*p*
Upregulation						
Q5M8C3	*Serpina4*	Serine (Or cysteine) proteinase inhibitor, clade A (Alpha-1 antiproteinase, antitrypsin), member 4	13.76	3.78	0.70	0.00
Q6YDN7	*Cdc26*	Anaphase-promoting complex subunit CDC26	7.80	2.96	0.81	0.03
Q5EB90	*Polr2c*	Polymerase (RNA) II	7.73	2.95	0.80	0.02
P23928	*Cryab*	Alpha-crystallin B chain	2.42	1.27	0.89	0.00
P10068	*Crygf*	Gamma-crystallin F	2.20	1.14	0.98	0.00
A0A0H2UHM3	*Hp*	Haptoglobin	2.17	1.12	0.89	0.00
P62697	*Crybb2*	Beta-crystallin B2	2.17	1.12	0.95	0.00
A0JN13	*Cryaa*	Alpha-crystallin A chain	2.14	1.10	0.86	0.00
Q8CGQ0	*Cryba2*	BetaA2-crystallin	2.12	1.08	0.96	0.04
P02767	*Ttr*	Transthyretin	2.11	1.08	0.86	0.01
P62329	*Tmsb4x*	Thymosin beta-4	2.07	1.05	0.78	0.00
P02524	*Crybb3*	Beta-crystallin B3	1.95	0.97	0.88	0.02
P02523	*Crybb1*	Beta-crystallin B1	1.94	0.96	0.87	0.02
Q8CHN7	*Pcp4*	Calmodulin regulator protein PCP4	1.92	0.94	0.86	0.00
P63249	*Pkia*	cAMP-dependent protein kinase inhibitor alpha	1.89	0.92	0.89	0.03
Q62785	*Pdap1*	28 kDa heat- and acid-stable phosphoprotein	1.86	0.90	0.77	0.02
P14881-2	*Cryba1*	Beta-crystallin A3	1.83	0.87	0.88	0.00
P02770	*Alb*	Albumin	1.81	0.86	0.85	0.02
B0K010	*Txndc17*	Thioredoxin domain-containing protein 17	1.81	0.86	0.70	0.02
A0A0G2JZ73	*Serpina1*	Alpha-1-antiproteinase	1.78	0.83	0.82	0.01
A0A0G2K376	*Fip1l1*	Pre-mRNA 3′-end-processing factor FIP1	1.75	0.80	0.70	0.01
Q6PDU6	*Hbb-b1*	Beta-glo	1.70	0.77	0.80	0.04
A0A1K0FUA6	*LOC100134871*	Globin a2	1.69	0.76	0.83	0.03
P14046	*A1i3*	Alpha-1-inhibitor 3	1.68	0.75	0.84	0.01
P20059	*Hpx*	Hemopexin	1.66	0.73	0.85	0.03
Q923W4	*Hdgfl3*	Hepatoma-derived growth factor-related protein 3	1.63	0.71	0.84	0.02
D3ZXP8	*Pcp2*	Purkinje cell protein 2	1.61	0.69	0.87	0.00
W4VSR7	*Try10*	RCG64520	1.59	0.67	0.84	0.01
P14841	*Cst3*	Cystatin-C	1.58	0.66	0.99	0.01
B4F758	*Hmgb1*	High mobility group protein B1	1.57	0.65	0.78	0.02
D4A9Z8	*Chmp4bl1*	Chromatin-modifying protein 4B-like 1	1.57	0.65	0.73	0.02
A0A0G2K6H5	*Cfdp1*	Craniofacial development protein 1	1.55	0.64	0.80	0.01
P47819	*Gfap*	Glial fibrillary acidic protein	1.55	0.63	0.85	0.01
Q05175	*Basp1*	Brain acid soluble protein 1	1.53	0.61	0.72	0.04
P26772	*Hspe1*	10 kDa heat shock protein, mitochondrial	1.51	0.59	0.79	0.00
A0A0G2JSW3	*Hbb*	Hemoglobin subunit beta-1	1.46	0.55	0.78	0.01
P11030	*Dbi*	Acyl-CoA-binding protein	1.46	0.54	0.86	0.04
Q4KLJ1	*Sfrs7*	RCG61762, isoform CRA_a	1.42	0.51	0.88	0.01
P31044	*Pebp1*	Phosphatidylethanolamine-binding protein 1	1.41	0.49	0.82	0.04
D4A6E3	*Mug1*	Murinoglobulin-1	1.38	0.47	0.78	0.02
F1LNF1	*Hnrnpa2b1*	Heterogeneous nuclear ribonucleoproteins A2/B1	1.38	0.47	0.84	0.03
D3ZM85	*Cplx4*	Complexin 4	1.38	0.46	0.71	0.02
P55053	*Fabp5*	Fatty acid-binding protein 5	1.37	0.45	0.73	0.01
Q6P6G9	*Hnrnpa1*	Heterogeneous nuclear ribonucleoprotein A1	1.36	0.45	0.77	0.01
D4ABK7	*Hnrnph3*	Heterogeneous nuclear ribonucleoprotein H3	1.36	0.45	0.78	0.02
P11240	*Cox5a*	Cytochrome c oxidase subunit 5A	1.36	0.45	0.79	0.02
B1WBQ0	*Cdc5l*	CDC5 cell division cycle 5-like	1.35	0.43	0.80	0.02
Downregulation						
P68370	*Tuba1a*	Tubulin alpha-1A chain	0.74	−0.43	0.99	0.02
D3ZC55	*Hspa12a*	Heat shock 70kDa protein 12A (Predicted), isoform CRA_a	0.73	−0.46	0.73	0.03
Q8CJD2	*N/A*	Guanylate cyclase	0.72	−0.47	0.81	0.03
F1M779	*Cltc*	Clathrin heavy chain	0.71	−0.49	0.87	0.00
C7C5T2	*Pfkp*	ATP-dependent 6-phosphofructokinase	0.69	−0.53	0.76	0.00
P01830	*Thy1*	Thy-1 membrane glycoprotein	0.67	−0.58	0.74	0.00
A0A0G2K0T6	*Sncg*	Gamma-synuclein	0.58	−0.78	0.74	0.01
D4AE00	“*Obsolete*”	“Obsolete”	0.55	−0.87	0.74	0.01
Q8K4V4	*Snx27*	Sorting nexin-27	0.53	−0.90	0.81	0.04
P12007	*Ivd*	Isovaleryl-CoA dehydrogenase, mitochondrial	0.50	−0.99	0.77	0.03
Q8CFN2	*Cdc42*	Cell division control protein 42 homolog	0.42	−1.25	0.80	0.01
P19527	*Nefl*	Neurofilament light polypeptide	0.38	−1.41	0.81	0.04
Q6EV70	*Pofut1*	GDP-fucose protein O-fucosyltransferase 1	0.22	−2.19	0.74	0.04
B1H267	*Snx5*	Sorting nexin-5	0.06	−3.95	0.78	0.00

**Table 5 ijms-24-12109-t005:** Meta-analysis of Gene Ontology (GO) for 1 wk, 4 wk, and 8 wk after pONT. The top 10 categories of biological processes, molecular function, and cellular components were shown based on the *p*-values in 8 wk. *p*-value < 0.05 of GO terms in three time points were shown in red. DE: differentially expressed; T/N: Temporal/Nasal.

	GO ID	T/N-1 wk		T/N-4 wk		T/N-8 wk	
		Genes (DE/All)	*p*	Genes (DE/All)	*p*	Genes (DE/All)	*p*
GO term-Biological process							
lens development in camera-type eye	GO:0002088	N/A	N/A	1/14	0.364	8/14	2.4 × 10^−6^
neuron projection regeneration	GO:0031102	N/A	N/A	2/16	0.089	8/16	9.1 × 10^−6^
response to wounding	GO:0009611	1/78	0.915	6/79	0.036	18/79	2.0 × 10^−5^
visual system development	GO:0150063	1/65	0.871	7/67	0.004	15/67	1.3 × 10^−4^
sensory system development	GO:0048880	1/65	0.871	7/67	0.004	15/67	1.3 × 10^−4^
eye development	GO:0001654	1/65	0.871	7/67	0.004	15/67	1.3 × 10^−4^
lipid biosynthetic process	GO:0008610	1/68	0.883	1/68	0.893	15/68	1.5 × 10^−4^
negative regulation of endopeptidase activity	GO:0010951	4/40	0.031	3/40	0.131	11/40	1.5 × 10^−4^
negative regulation of peptidase activity	GO:0010466	4/41	0.034	3/41	0.139	11/41	2.0 × 10^−4^
negative regulation of hydrolase activity	GO:0051346	4/54	0.079	4/55	0.094	13/55	2.1 × 10^−4^
GO terms-molecular function							
structural constituent of eye lens	GO:0005212	N/A	N/A	1/12	0.325	8/12	2.9 × 10^−7^
structural molecule activity	GO:0005198	7/111	0.054	6/112	0.145	23/112	2.7 × 10^−6^
serine-type endopeptidase inhibitor activity	GO:0004867	2/13	0.060	3/13	0.007	6/13	1.6 × 10^−4^
endopeptidase inhibitor activity	GO:0004866	4/25	0.007	3/25	0.044	8/25	2.7 × 10^−4^
peptidase inhibitor activity	GO:0030414	4/25	0.007	3/25	0.044	8/25	2.7 × 10^−4^
endopeptidase regulator activity	GO:0061135	4/28	0.010	3/28	0.058	8/28	6.4 × 10^−4^
signaling receptor regulator activity	GO:0030545	N/A	N/A	1/18	0.445	6/18	0.001
peptidase regulator activity	GO:0061134	4/31	0.014	3/31	0.075	8/31	0.001
monosaccharide binding	GO:0048029	1/20	0.472	1/20	0.481	6/20	0.002
receptor ligand activity	GO:0048018	N/A	N/A	N/A	N/A	5/15	0.003
GO terms-cellular component							
extracellular space	GO:0005615	7/125	0.079	9/127	0.014	28/127	1.1 × 10^−7^
extracellular region	GO:0005576	9/163	0.052	12/165	0.004	30/165	3.1 × 10^−6^
hemoglobin complex	GO:0005833	N/A	N/A	1/4	0.118	3/4	0.002
haptoglobin-hemoglobin complex	GO:0031838	N/A	N/A	N/A	N/A	3/4	0.002
neurofilament	GO:0005883	N/A	N/A	1/5	0.146	3/5	0.004
presynaptic intermediate filament cytoskeleton	GO:0099182	N/A	N/A	1/2	0.061	2/2	0.006
intermediate filament	GO:0005882	1/19	0.444	4/19	0.002	5/19	0.013
axon	GO:0030424	7/209	0.449	7/211	0.486	25/211	0.017
postsynaptic intermediate filament cytoskeleton	GO:0099160	N/A	N/A	1/3	0.090	2/3	0.017
growth cone membrane	GO:0032584	N/A	N/A	N/A	N/A	2/3	0.017

## Data Availability

The data presented in this study are available in this article.

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
