# Peer review of "Differential Responses of Retinal Neurons and Glia Revealed via Proteomic Analysis on Primary and Secondary Retinal Ganglion Cell Degeneration"

_ijms, 2023, doi:10.3390/ijms241512109_

Round 1

Reviewer 1 Report

The authors previously showed that temporal partial optic injury resulted in early temporal RGC loss at 1 week and late nasal RGC loss at 8 weeks (reference # 23). Using this model, the present study compared several proteins between temporal and nasal area in retinal samples. Among the significant changed proteins, the authors performed immunohistochemial analyses of gamma-synuclein, GFAP, S100, and ALDH1A1. The findings are potential interesting. However, some concerns need to be addressed.

Major

Figure 6. This is 8 weeks after pONT. However, there are a lot of RBPMS positive cells in both the nasal and temporal area. Presumably, 30 % and 50 % of RBPMS positive RGCs may remain at temporal and nasal area at 8 weeks, respectively (reference #23). Therefore, the authors should replace with better photos. Also, the authors need to show earlier time points (1 week and 4 weeks) in this regard. These comparisons may be convincing for the readers.

Minor

Figure 5. Please indicated Control, 1 week, 4 weeks, and 8 weeks at upper portion like figure 7.

Figure 6. Left side “SNUG” should be “SNCG”.

Figure 9. If the authors can show double staining of ALDH1A1 and HO-1 or SOD, the antioxidant response can be reasonable.

Reviewer 2 Report

The authors compare the temporal (1, 4, and 8 week) retinal proteome in the rat following temporal partial optic nerve transection and report the significant differences between the concentrations in the temporal versus the nasal retina at each of those timepoints.

The paper in general is well-written.

1, In the abstract and in the manuscript, make clearer that the overall retinal concentration of ALDH1A1 increased – but appeared to decrease in the RNFL by IHC.

2. line 160  Change regulated to “up-regulated”

3. Figure 4. Explain exactly what happened to prevent the points of Albumin of 0.97 at 1 week and Beta-glo at 1.76 at 1 week from being designated as significant.

4. Figure 5,

Appears different magnifications are used in this figure – especially for the bottom temporal retina row – needs correction.  Also appears loss in the outer retina is significant compared to the presumed left-hand control.  Explain the decrease in the other layers.

5, The authors report that GFAP increased more temporally than nasally molecularly at 4 and 8 weeks. However, figure 8 shows more intense GFAP and S100 staining of the NASAL tissue specimen than the Temporal specimen at 8 weeks.  Explain

Also. Difficult to discern the processes indicated by the arrows.

6.  In a prior paper, the authors examined controls and the temporal and nasal retinas at 2 weeks after pONT. Secondary Retinal Ganglion Cell Degeneration Identified by Integrated SWATH and Target-Based Proteomics

Why was aldehyde dehydrogenase 1A1 - ALDH1A1 not mentioned/found in the results of that study at 2 weeks since the authors found a significant differential increase of temporal versus nasal ALDH1A1 at 1 and 4 weeks in this study? 

.

Round 2

Reviewer 2 Report

Manuscript is improved